# How Long Is Long Enough? Controlling for Acute Caffeine Intake in Cardiovascular Research

**DOI:** 10.3390/brainsci13020224

**Published:** 2023-01-29

**Authors:** Shara S. Grant, Kye Kim, Bruce H. Friedman

**Affiliations:** 1Department of Psychology, Virginia Tech, 109 Williams Hall, Blacksburg, VA 24061, USA; 2Department of Pharmacology and Physiology, George Washington University, 2300 I Street NW Ste 640, Washington, DC 20037, USA; 3Department of Psychiatry and Behavioral Medicine, Virginia Tech Carilion Clinic School of Medicine, 1 Riverside Circle, Roanoke, VA 24016, USA

**Keywords:** caffeine, coffee, cardiovascular, methodology, autonomic

## Abstract

Caffeine substantially affects cardiovascular functioning, yet wide variability exists in caffeine control procedures in cardiovascular reactivity research. This study was conducted in order to identify a minimal abstention duration in habitual coffee consumers whereby cardiovascular reactivity is unconfounded by caffeine; Six hours (caffeine’s average half-life) was hypothesized. Thirty-nine subjects (mean age: 20.9; 20 women) completed a repeated measures study involving hand cold pressor (CP) and memory tasks. Caffeinated and decaffeinated coffee were administered. The following cardiovascular indices were acquired during pre-task, task, and post-task epochs prior to coffee intake, 30 min-, and six hours post-intake: Heart rate (HR), high-frequency heart rate variability (HF-HRV), root mean squared successive differences (RMSSD), systolic and diastolic blood pressures (SBP, DBP), mean arterial pressure (MAP), pre-ejection period (PEP), left ventricular ejection time (LVET), systemic vascular resistance (SVR), systemic vascular resistance index (SVRI). Results support the adequacy of a six-hour abstention in controlling for caffeine-elicited cardiovascular changes. The current study offers a suggested guideline for caffeine abstention duration in cardiovascular research in psychophysiology. Consistent practice in caffeine abstention protocols would promote validity and reliability across such studies.

## 1. Introduction

Caffeine, the most widely used psychoactive substance worldwide, modulates myriad physiological, behavioral, and cognitive processes [1]. In particular, caffeine has well documented effects on cardiovascular variables, e.g., [2,3,4,5]. Caffeine use is particularly high among college students, who often comprise the subject samples in psychophysiological studies. As such, caffeine presents a potential confound in psychophysiological research focused on cardiovascular activity. Yet, there currently is no standard for caffeine abstention time prior to participation in such studies [6]. Indeed, wide variability exists in these practices.

In this paper, we describe a cardiovascular study conducted with the aim of moving the field toward more uniform protocols for caffeine abstention. The present investigation may be viewed as a “proof of concept” study addressing the need for additional empirical studies of this issue. In other words, this study implements a method to demonstrate the feasibility of a six-hour abstention period in reducing effects of caffeine on variables of interest [7]. First, we provide an overview of the physiological effects of caffeine. We then explicate the strategy behind the study’s design, which involved assessment of cardiovascular reactivity to lab manipulations in a sample of healthy young adults who were regular moderate consumers of caffeine. The study’s methodology and results are then detailed, followed by a discussion of the implications of the findings for standardizing caffeine abstention protocols.

### 1.1. The Physiology of Caffeine

Caffeine is a fully bioavailable drug that is absorbed by the small intestine within approximately 30–45 min after ingestion and has an average elimination half-life of six hours [8,9]. Dosage as well as diverse inter-individual factors (e.g., diet, nicotine, oral contraceptives, body mass index, and genetics) affect caffeine metabolism and physiological effects of the drug [10,11,12]. The key contributor of caffeine’s biological effects is antagonism of A1 and A2 subtypes of the adenosine receptor, which causes primarily sympathomimetic effects [8,13]. The present paper describes a study of the effects of the drug at moderate or dietary doses.

Caffeine exerts numerous effects on tonic cardiovascular activity as well as additive effects during cardiovascular reactivity to acute physical and mental stress, e.g., [3,13]. Acute cardiovascular effects of caffeine commonly include significant transient post-consumption elevations in resting systolic and diastolic blood pressure (SBP; DBP) among habitual and non-habitual consumers [14,15,16]. In response to caffeine-induced pressor increases, baroreflex responses mediate HR activity in a compensatory fashion [14]. Studies typically report either small decreases, e.g., [13] or no change in resting heart rate (HR) with acute consumption of moderate doses of caffeine, e.g., [17]. Several investigations have explored and confirmed caffeine’s additive influence on acute cardiovascular reactivity to both mental and physical laboratory stressors, particularly regarding BP and HR responses. Studies show exaggerated acute BP reactivity to caffeine versus placebo administration, e.g., [3,4,5]. This additional cardiovascular reactivity to caffeine occurring during acute stress appears to be associated with neuroendocrine upregulation (specifically potentiation of epinephrine reactivity due to caffeine-caused elevations in resting plasma epinephrine) [3].

Typically, no drug-induced changes are shown in systolic time intervals pre-ejection period (PEP) and left-ventricular-ejection time (LVET), e.g., [18]. Findings using male samples have shown resting pressor increases associated with acute caffeine intake are due to elevations in systemic vascular resistance (SVR), with no change in cardiac output [15]. However, the mechanisms driving acute post-caffeine elevations in BP reactivity to laboratory stressors may differ by sex. In one study, despite similar degrees of BP reactivity to mental stress for men and women following caffeine consumption, SVR increases were significantly greater among men, whereas for women, pressor changes were associated with increased cardiac output [19]. Further research must examine potential sex differences in caffeine-elicited changes in vascular activity [20].

Effects of acute caffeine intake on heart rate variability are relatively understudied. High-frequency heart rate variability (HF-HRV) reflects vagal activity and broadly, *autonomic flexibility*, which positively relates to cardiovascular and emotional health [21,22]. Studies within a review of caffeine and HRV generally reported caffeine-induced increases in resting HF-HRV [2]. Of note, caffeine’s acute cardiovascular effects are complex and caffeine produces antagonistic actions culminating in net effects on autonomic indices. 

Despite common belief in the habituation of caffeine’s effects among regular consumers, extensive research demonstrates insufficient tolerance to caffeine’s cardiovascular effects [16,23]. Additionally, while most extant caffeine-related cardiovascular research focuses on acute effects, a review of caffeine withdrawal suggests associations of caffeine abstinence with increased cerebral blood velocity and decreased BP [24]. Accordingly, acute decreases in cerebral blood flow accompany caffeine intake [25]. Furthermore, protracted or withdrawal effects of caffeine are widely known to influence cognitive variables and affective factors such as mood. For example, a great portion of studies in a review concerning caffeine withdrawal indicated impaired cognitive performance (48%) irritability (35%) and lack of motivation (50%) following abrupt caffeine abstinence among regular consumers [24].

Caffeine’s acute and extended effects on physiological and psychological variables likely impact reliability and validity of research outcomes. Our review of common control procedures for caffeine abstention among recently published peer-reviewed studies of cardiovascular reactivity found that only 22% of the studies explicitly stated pre-experimental caffeine abstention [6]. Hence, it is unclear how or whether potential effects of caffeine were accounted for among the remaining investigations. Caffeine abstention durations among the total identified studies ranged from 1 to 24 h (*SD* = 8.02), signifying considerable variation. Widely variable practices in caffeine control both within and across cardiovascular studies may lead to potentially confounded results and difficulty in cross-comparisons of research findings. No standardized guidelines exist for caffeine abstention prior to studies of cardiovascular reactivity. The impetus for the current study was rooted in evidence of this apparent methodological deficiency.

This concern extends beyond caffeine-related research, and to physiological research more broadly; caffeine’s effects could modulate a variety of both resting and reactivity indices. A clear need exists for systematic study of optimal abstention durations that balance the need to reduce acute caffeine effects on cardiovascular reactivity along with the minimization of deprivation effects.

This research gap is especially salient considering the ubiquitous consumption of caffeine, particularly among college students who are often the subjects in psychophysiological research. Caffeine consumption among youth has increased over the past decade, with individuals aged 18–24 consuming an average of nearly 288 mg of caffeine daily [1,26]. A survey of caffeine consumption in a large sample drawn from five geographically distributed U.S. universities found that 92% of students reported caffeine use in the past year, the primary source of which was coffee [27]. These figures closely mirror data in the general U.S. population [28].

### 1.2. Aims and Objectives

The primary aim of the present investigation was to identify an optimal duration for participant caffeine abstention whereby the drug minimally affects cardiovascular activity, particularly regarding reactivity to lab manipulations. A six-hour duration was hypothesized, based on the average elimination half-life of the drug. The selection of the drug’s average half-life was based on a desire to avert reactivity effects and potential deprivation effects on physiological and psychological variables of interest. Caffeine’s effects depend on several intervening factors, so the present study pertained specifically to a young healthy sample of moderate regular caffeine consumers. This is the first known study to systematically investigate effects of caffeine intake with the primary purpose of identifying ideal pre-experimental abstention durations in cardiovascular reactivity research.

Given potentially additive effects of caffeine on stress reactivity (e.g., [3]), examining cardiovascular reactivity to stress inductions was of particular interest. Using a repeated measures design, comparison of cardiovascular reactivity to two tasks (hand cold pressor (CP) and working memory) was made across three experimental phases: *Phase 1*: pre-caffeinated; and *Phases 2* and *3*, respectively: 30 min- and six hours-following consumption of caffeinated coffee.

The study employed repeated measures analyses of mean, reactivity, and recovery values of HR, SBP, DBP, mean arterial pressure (MAP), HF-HRV, root mean square of successive R-R interval differences (RMSSD), PEP, LVET, SVR, and systemic vascular resistance index (SVRI). No significant differences in cardiovascular activity between Phases 1 and 3 were predicted to emerge, thereby suggesting the adequacy of a six-hour duration in controlling for caffeine-elicited cardiovascular changes. Variables hypothesized to be associated with acute caffeine intake were mean resting and reactivity increases in SBP, DBP, MAP, SVR, SVRI, RMSSD, and HF-HRV, decreases in HR, and no change in PEP or LVET.

## 2. Methods

### 2.1. Subjects and Study Design

Based on G*Power software’s analysis, ability to attain a medium effect size (0.3) for the repeated measures analyses with a 0.95 power value at α = 0.05 requires 40 subjects. Forty-three subjects were recruited from Virginia Tech and the nearby community; of which 39 yielded useable data. Hence, 39 subjects (M_age_ = 20.9 years, SD = 1.9; Range_age_ = 18–24 years; *n* = 20 women) completed the study. Subjects were screened via self-report questionnaires to ensure the following criteria for participation were met. The sample was comprised of right-handed healthy regular consumers of self-reported moderate amounts of caffeine (average two to three cups of coffee daily; approximately 200–300 mg/day [29,30]. Subjects completed a detailed journal documenting their daily intake of all caffeine-containing beverages and foods for one week leading up to the first session, as well as on the days in between experimental days). Exclusionary criteria were history of cardiovascular disease, metabolic disease, neurological, respiratory, or psychiatric disorder, history of traumatic brain injury or recent history of brain injury, smoking within the past year, current use of medications with known cardiovascular effects, and use of prescription medications taken less than one month prior to the study. Subjects were monetarily compensated for study participation. This study was approved by the Virginia Tech Institutional Review Board. Informed consent was obtained from all subjects involved in the study. Refer to Table 1 for descriptive statistics for the sample. 

A fully within-subjects design was employed. Each subject completed a total of four laboratory sessions occurring on two days: two morning sessions lasting 1 h and 45 min in duration and two afternoon sessions lasting 30 min each. Prior to each morning session, subjects were instructed to abstain from consumption of alcohol within 24 h, from caffeine within 12 h, and to sleep for approximately eight hours overnight. Instruction also included refraining from eating or vigorously exercising within two hours preceding each session. Instructional adherence was confirmed via self-report on recent health behavior questionnaires. Between the first and second days of the study, instructions were to resume normal caffeine consumption until the abstention period prior to the second day. 

### 2.2. Procedure

#### 2.2.1. Phase One

On the first experimental day, subjects entered the laboratory at 8:00 a.m. or 10:00 a.m. Following informed consent, physiological recording equipment for ICG, electrocardiography (ECG), and respiration was attached to subjects. Subsequently, subjects sat before a computer to complete a health screening questionnaire and the Brief Mood Inventory Scale (BMIS) [31]. A cuff was then placed on the left wrist for semi-continuous BP measurements. Recording of all physiological measures continued simultaneously throughout the session. Following three minutes of additional resting to further acclimate to the experimental environment, subjects began a series of two successive three-minute pre-task, task, and post-task epochs. 

Resting pre-task measures were recorded prior to completion of one of two stated tasks (working memory task or cold pressor task), which again were counterbalanced across subjects and sessions. During pre-task periods, subjects viewed non-stimulating videos featuring aquatic life [32]. Pre-task periods were employed for the collection of resting physiological measures [33]. Subjects watched engaging but non-stimulating videos, which were muted to prevent potential confounding reactivity effects stemming from responses to music. Six different segments of videos displaying marine life were viewed during all three-minute pre-task epochs [32]. Following completion of questionnaires after coffee consumption, a video of scenes portraying everyday life in various cultures was shown for the remainder of the 30 min period [34]. All video clips were counterbalanced across subjects and sessions.

For the CP task, subjects immersed the right hand in cold water (6 to 8 °C) up to the wrist for the duration of the epoch while keeping the hand relatively still. For the memory task, each trial involved presentation of four slides (See Figure 1). On the first slide, a small white fixation cross appeared against a black background on a laptop computer screen for 750 ms. This preceded a series of four white capital nonsense letters presented for 550 ms. On the following slide, the letter series appeared with a single blue capital letter of the same size above the masked series for 550 ms. On the final slide of the trial, subjects viewed a single blue letter in isolation in the center of the screen for a maximum of 2550 ms. At this point, subjects pressed a key indicating whether or not the blue letter was presented in the previously shown series of letters (the number 1 indicating ‘yes’ or 2 indicating ‘no’). On certain trials (masking trials) this letter was the masking letter. On other trials, this letter was indeed present in the letter string. Trials were presented randomly. If no response was given in the allotted durations, the next slide was advanced. During the first session only, subjects completed a series of 14 practice trials prior to completing 44 trials of the task. Prior to initiation of each memory task, individuals were encouraged to perform optimally well (90% or greater across sessions) in order to receive an additional monetary reward at the end of the fourth session. Immediately subsequent to the post-task period, subjects provided task ratings via pen and paper. Each task period was followed by a three-minute post-task period in which the subject sat quietly. This completed Phase 1 of the study.

#### 2.2.2. Phase Two

Subjects then consumed eight ounces of either hot caffeinated coffee (230 mg caffeine) or an identical amount of hot decaffeinated coffee (5 mg caffeine) depending on the condition assigned. Caffeine was decidedly administered via caffeinated coffee (230 mg) and decaffeinated coffee (5 mg) because in the US, approximately 46% of caffeine consuming youth ages 18–24 regularly consume coffee [1]. Coffee was brewed via a standard Keurig cup coffee maker. Subjects were offered a maximum of two small standard sized packets of creamer and sugar or artificial sweetener to add to the coffee, as consuming coffee without cream or sweetener may have produced an aversive reaction among those who normally do not consume coffee this way. No consistent evidence reveals systematic acute effects of substances in creamer or sugar of the named amounts on variables of interest in this study [37,38].

Caffeine conditions were counterbalanced across morning sessions and subjects were blind to the condition assigned. Subjects were allowed a maximum of five minutes to consume the beverage in its entirety. Once consumed, researchers allowed exactly 30 min following consumption for the caffeine to take effect. During this period, individuals completed the Depression Anxiety Stress Scale (DASS) [39] prior to watching another non-stimulating video [34,40] for the remainder of the 30 min period. Subjects then briefly completed the BMIS once again to assess possible mood changes associated with acute reactivity to caffeine. The end of this period initiated completion of another pre-task, task, and post-task sequence, involving both tasks as outlined, in order to assess acute physiological reactivity to caffeine. This completed Phase 2 of the study. At the end of the session, researchers obtained height and weight measures in order to derive body mass index (BMI). Subjects were reminded to return to the lab for the afternoon session, which began six hours following coffee consumption. Subjects were also reminded to abstain from consuming caffeine from any source during this period, and not to consume food or beverages nor engage in vigorous physical activity within two hours of the afternoon session. Researchers requested a brief, general, hour-by-hour account of activities engaged in during the interim period. None of the subjects reported consuming caffeine during this period.

#### 2.2.3. Phase Three

Subjects returned to the lab in the afternoon (six hours post-coffee consumption) to complete Phase 3 of the study. During the afternoon session, the study procedure mirrored the morning procedure, excluding coffee consumption. Following completion of the Recent Health History Questionnaire, identical physiological measures were collected during completion of only one pre-task, task, and post-task sequence that included both tasks. Tasks and pre-task videos were again counterbalanced across subjects and sessions. At the end of the afternoon session, subjects were instructed to resume typical consumption of caffeinated beverages until the abstention period prior to the next morning session, as well as to continue completing the caffeine consumption log. Monetary compensation for completion of the first two sessions was given at this time.

On a separate day during the following week, subjects returned to the lab at 8:00 a.m. or 10:00 a.m. to complete the third and fourth experimental sessions. Most subjects returned on the same day of the week as in the first session, during the same time slot, although this was not the case for each subject, due to scheduling variations. This procedure was nearly identical to that of the first day, but in the opposing caffeine condition. Additionally, the memory task was not preceded by practice trials on the second day. Researchers debriefed subjects at the final session (session four) before subjects received monetary compensation for the final two sessions.

### 2.3. Measures (Refer to Table 2 for List of Measures Used)

#### 2.3.1. Physiological Measures

All physiological signals were acquired using Biopac Systems, Inc. MP150 hardware (Biopac Systems Inc., Goleta, CA, USA). The MP150 digitally sampled signals at 1000 Hz. Physiological data were acquired using Biopac AcqKnowledge 4.4 software in order to detect and remove movement artifact. ConMed Suretrace conductive Ag/AgCl adhesive pre-gelled electrodes were used for collection of ICG and ECG data. All physiological measures were collected in real time, then separately averaged within pre-task, task, and post-task epochs prior to statistical analyses. 

##### Electrocardiography (ECG)

The ECG signal was amplified through the ECG100C system and measured using a standard lead II electrode configuration. A detector system computed beat-to-beat distances between peaks of the QRS (R-waves) in order to calculate interbeat interval (IBI) values. IBI’s were then imported into Kubios software (Version 2.2, 2004, University of Eastern Finland, Kuopio, Finland) for HR and HRV measures. HF-HRV and RMSSD were, respectively regarded as frequency-based and time-based proxies of vagally mediated cardiac control. Values were obtained using spectral analysis of this signal and yielded variability in the high frequency spectrum corresponding to respiration (0.12–0.40 Hz). Specifically, a Fast-Fourier Transform function was performed for the IBI time series in order to derive power (ms^2^) in this frequency band. RMSSD and HF-HRV calculations were converted with a natural logarithm transformation prior to statistical analyses. 

##### Respiration

Respiration rate values were obtained to control for possible respiratory effects on HRV parameters. A Biopac respiration belt with TP-TSD201 Respiratory Effort transducer was wrapped around the subject at the sternum and fastened close but comfortably [41]. The respiration waveform was resampled at 62.5 samples per second and transformed using a digital band pass filter with low- and high-frequency cutoffs fixed at 0.05 and 1.0 Hz, respectively. The waveform was fixed at 5000 coefficients, based on guidelines suggesting number of coefficients at 4× (Waveform Sampling Rate/Lowest Frequency Cutoff for Filter; (Biopac Systems Inc., AcqKnowledge Software Guide, Goleta, CA, USA).

##### Blood Pressure (BP)

The Fusion Noninvasive Semi-Continuous Blood Pressure Measurement System Unit was used for collection of the blood pressure signal (Medwave, Inc., Danvers, MA, USA). The monitor included a sensor contained in a Velcro wrist strap, which was placed on the subject’s left wrist atop the radial artery. The device sampled both SBP and DBP values four times per minute. The signal was amplified by the Biopac NIBP100B system. Mean Arterial Pressure (MAP) values were equal to (2 × DBP) + SBP/3. SBP and DBP were separately averaged within each epoch prior to performing MAP calculations. Epochs containing greater than 45 consecutive seconds (approximately 3 samples) of an absent signal were treated as missing data.

##### Impedance Cardiography (ICG)

Noninvasive thoracic impedance cardiography (ICG) was recorded via standard tetrapolar spot electrode placement along the spine of each subject. One electrode was placed at the top of the neck, and a second electrode was placed roughly at the seventh vertebra of the cervical spine (approximately 5 cm from the first electrode). A distance of 25–35 cm (depending on subject height) was between the second and third electrodes from the uppermost electrode. The Biopac EBI100C system amplified the signals. Delta Z was derived using Acqknowledge 4.4. The software extracted the dZ/dt waveform, providing the following: C-point location using adaptive template matching, X-point location using the minimum dZ/dt 150–275 ms after the C-point, and B-point location using the minimum derivative in the C-QRS interval. Two independent scorers visually detected the B point. Each dZ/dt waveform was visually inspected twice by two researchers for accuracy and manually edited where necessary. Subjects’ height, weight, and distance between second and third electrodes were entered into the analysis routine. The Kubicek/Rho method for stroke volume calculation was selected. 

#### 2.3.2. Psychological Measures

##### Depression Anxiety and Stress Scale (DASS)

The DASS was used to broadly explore associations between depressive, anxious, and stress-related emotions and physiological responses to caffeine. The 42-item self-report instrument was designed to index negative emotional states characteristic of depression, anxiety and stress/tension [39]. Each subscale contains 14 items. Items on the Depression scale assess dysphoria, devaluation of life, hopelessness, self-deprecation, anhedonia, lack of interest or involvement in daily activities, and inertia. The Anxiety scale indexes autonomic arousal, skeletal muscle effects, situational anxiety, and the subjective experience of anxious affect. The Stress scale measures chronic non-specific arousal (i.e., nervous arousal, difficulty relaxing, and irritability or impatience). Four-point rating scales evaluate frequency and extent to which each emotional state was experienced over the past week. Exploratory and confirmatory factor analyses show satisfactory internal consistency among all scales (α~ = 0.96, 0.89 and 0.93 for Depression, Anxiety, and Stress, respectively) [39,42]. 

##### Brief Mood Introspection Scale (BMIS)

This brief self-report state measure assessed current mood states during each experimental phase. The BMIS contains 16 items, each on a four-point Likert scale, indicating how well each adjective described one’s present mood. The scale includes four dimensions of mood: Pleasant-Unpleasant, Arousal-Calm, Positive-Tired, and Negative-Calm. Greater numerical responses for each dimension are associated with greater scores (e.g., higher scores for “Pleasant” or “Arousal” correspond with greater self-rated pleasantness or arousal). The BMIS has good factor validity and Cronbach’s alpha reliabilities ranging from 0.76 to 0.83 [31].

##### Task Ratings

Subjects provided Likert scale ratings based on perceived difficulty of each memory task performed as well as pain level experienced during each CP task. Ratings were completed once at each experimental phase following post-task periods. Memory task difficulty ratings ranged from 1 (very easy) to 9 (very difficult). Ratings for the CP task ranged from 0 (no pain) to 10 (worst possible pain) (National Institute of Clinical Studies, 2011).

### 2.4. Experimental Tasks

The working memory task was selected to index primarily beta-adrenergic influences, and the CP task was chose to reflect alpha-adrenergic cardiovascular activity. An adaptation of Sternberg’s original working memory task was employed [35,36]. The task was created using E-Prime 2.0 stimulus presentation software (Psychology Software Tools, Inc., Sharpsburg, PA, USA). The CP task entailed using a standard water cooler filled with water 6 to 8° Celcius, a temperature range which sufficiently and reliably elicits increased BP through vasoconstriction [43]. A water circulator attached to an interior wall of the cooler provided movement of the water to promote stable water temperature surrounding the hand. Tasks were counterbalanced across subjects and sessions. (Refer to Table 2 for list of tasks used). 

**Table 2 brainsci-13-00224-t002:** Measures and tasks.

Physiological	Psychological/Self-Report	Experimental Tasks
**ECG-derived:** Heart Rate, High-Frequency Heart Rate Variability (HF-HRV), root mean square of successive R-R interval differences (RMSSD)	Depression Anxiety and Stress Scale (DASS): Depression, Anxiety, and Stress subscales Brief Mood Introspection Scale (BMIS): *Dimensions of Mood:* Pleasant-Unpleasant, Arousal-Calm, Positive-Tired, and Negative-Calm Self-Reported Task Ratings	Hand Cold Pressor (3 min) Working Memory Task (3 min): Percentage Correct
**Blood Pressure:** Systolic Blood Pressure (SBP), Diastolic Blood Pressure (DBP), Mean Arterial Pressure (MAP)		
**Impedance Cardiography (ICG)-derived:** Pre-ejection period (PEP), Left-ventricular ejection time (LVET), Systemic Vascular Resistance (SVR), and Systemic Vascular Resistance Index (SVRI)		

### 2.5. Data Reduction and Analyses

All statistical analyses were performed using the Statistical Package for Social Sciences version 24.0 for Windows statistical software (SPSS Inc., Chicago, IL, USA).

#### 2.5.1. Physiological Measures

Prior to performing statistical analyses, measures from each epoch were averaged separately, with data expressed as mean values for each physiological variable. Separate four-factor repeated measures Multivariate Analysis of Variance (MANOVA) tests for each physiological measure were used to examine the presence of significant within-subject mean differences based on four factors of interest: *Condition* (caffeine, decaffeinated), *Experimental Phase* (1, 2, 3), *Task* (memory, CP), and *Epoch* (pre-task, task, post-task). Experimental phases were as follows: (Phase 1) measures taken prior to consumption of coffee, (Phase 2) acute measures collected 30 min following intake of coffee, and (Phase 3) measures collected six hours following consumption of coffee. Repeated measures ANOVA tests were not selected for analyses due to stricter assumptions of the test.

Repeated measures MANOVA’s using within-subject physiological change scores from pre-task to task assessed physiological reactivity. Reactivity scores were calculated by subtracting mean pre-task from mean task values. In order to index physiological recovery, mean post-task scores were subtracted from pre-task scores in order to assess the extent to which subjects’ physiological measures returned or pre-task values within the allotted three-minute post-task period. MANOVA tests for reactivity and recovery were performed using factors *Condition*, *Task*, and *Experimental Phase*. BMI and oral contraceptive use (treated as a coded variable (1: use; 0: no use)) were analyzed as covariates. A Bonferonni confidence interval adjustment was used for all MANOVA analyses. Post hoc pairwise comparison analyses were also performed to further explore significant interactions and relationships. 

#### 2.5.2. Self-Report Measures

Analyses of DASS data were completed using a univariate repeated measures ANOVA tests, with Condition (Caffeine, Decaffeinated) as a factor in examination of possible associations of condition with self-reported perceptions of depression, anxiety, and stress. Analysis of BMIS responses was performed using a 3 (phase) × 2 (condition) repeated measures MANOVA. Self-reported difficulty and pain ratings for tasks were analyzed using an identical analysis. BMIS responses were scored on the following dimensions: Pleasant-Unpleasant and Arousal-Calm. Items included in the former dimension were: Active, Calm, Caring, Content, Happy, Lively, Loving, Peppy, Drowsy, Fed up, Gloomy, Grouchy, Jittery, Nervous, Sad, and Tired. Items included in the latter dimension were: Active, Caring, Fed up, Gloomy, Jittery, Lively, Loving, Nervous, Peppy, Sad, Calm, and Tired. Unpleasant and Calm items were reverse scored.

#### 2.5.3. Sex Effects

Finally, ancillary multivariate ANOVAs were performed to assess sex effects on mean, reactivity, and recovery values for each physiological measure. Sex differences were also assessed for mood via ANOVA tests for self-report responses on the caffeinated day. Again, BMI and contraceptive use were accounted for in the models as covariates. Independent t-tests for each physiological measure examined possible sex differences in cardiovascular task reactivity during the acute caffeinated phase. To assess hemodynamic forces associated with acute caffeinated reactivity to tasks, partial correlations were employed to test relationships between BP reactivity (SBP and DBP) and SVRI reactivity for male and female subjects, respectively, in order to explore possible mechanistic sex differences in cardiovascular reactivity following acute caffeine intake. BMI and contraceptive use were controlled for women, where for men, BMI was controlled for.

#### 2.5.4. Missing Data

Four of 43 recruited subjects were not included in the final sample due to attrition. Certain data from the remaining sample of 39 were not included in analyses due to equipment malfunction or confounding effects that appeared during the study. Specifically, three subjects were excluded from HR and HRV analyses, and two were not included in the BP analyses. Missing data were imputed using multiple imputation. A maximum of 4% of data for each dependent measure was absent prior to imputation. Seven iterations were used based on accepted recommendations [44]. Fifteen subjects were excluded from the ICG analyses due to equipment failure and/or noisy data for one or more of the subjects’ four sessions. As such, 24 subjects were included in ICG analyses. To address sphericity violations, Greenhouse-Geisser epsilon corrections (rather than multivariate (MANOVA) results) were used due to remaining small sample size following elimination of data from the fifteen subjects.

## 3. Results

See (a) Table 1 for descriptive statistics for demographic and physiological measures of the sample; (b) Table 3 for mean physiological values during the acute caffeine phase by task, and (c) Table 4 for mean physiological reactivity values by phase and task.

### 3.1. Heart Rate (HR)

A repeated measures MANOVA for mean HR showed a significant Condition × Phase interaction with greatest reduction in HR at phase 2 in the Caffeine condition [Wilks λ = 0.76, *F*(2,32) = 5.03, *p* = 0.013] (See Figure 2). A significant Condition × Task interaction displayed lower HR values for the memory task versus the CP task, but only in the caffeine condition [Wilks λ = 0.88, *F*(2,32) = 4.35, *p* = 0.045]. A pairwise comparison test showed significant differences between Phases 1 and 2 (*p* < 0.000), and 2 and 3 (*p* < 0.000), but no significant difference between Phases 1 and 3 (*p* = 0.88). 

Significant main effects occurred for HR reactivity, both for Condition and for Phase [Wilks λ = 0.74, *F*(1,33) = 11.36, *p* = 0.002; Wilks λ = 0.80, *F*(2,32) = 4.13, *p* = 0.025, respectively] (see Figure 3). Greatest HR reactivity occurred in the caffeine condition, specifically at Phase 2. Collapsed across conditions, a significant Phase × Task interaction was found, where lowest reactivity occurred at Phase 2 for the memory task (+3.06 BPM), and greatest reactivity was at Phase 2 for the CP task (+4.79 BPM) [Wilks λ = 0.75, *F*(2,32) = 5.31, *p* = 0.010]. 

### 3.2. High-Frequency Heart Rate Variability (HF-HRV)

A repeated measures MANOVA test of mean values did not reveal significant effects for HF-HRV values for Phase. However, significant interactions for Condition × Task, Condition × Epoch, and Task × Epoch appeared [Wilks λ = 0.78, *F*(1,33) = 9.71, *p* = 0.004; Wilks λ = 0.74, *F*(2,32), *p* = 0.008; Wilks λ = 0.79, *F*(2,32) = 4.26, *p* = 0.023, respectively]. Greater mean HF-HRV was found for the CP task than the memory task, but only on the caffeinated day. Mean HF-HRV values were greatest during tasks, and the caffeine condition was associated with greater mean values overall, compared to the decaffeinated condition. A Condition × Phase × Task interaction showed overall greatest mean HF-HRV during Phase 2, especially on the caffeinated day [Wilks λ = 0.79, *F*(2,32) = 4.28, *p* = 0.023]. A significant main effect of Condition was found for HF-HRV reactivity, where greatest acute task-elicited increases occurred on the caffeinated day [Wilks λ = 0.76, *F*(1,33) = 10.56, *p* = 0.003]. 

### 3.3. Root Mean Squared Successive Differences of R-R Intervals (RMSSD)

A repeated measures MANOVA test for mean RMSSD values revealed significant two-way interactions for Condition × Phase [Wilks λ = 0.83, *F*(2,32) = 3.37, *p* = 0.047] (see Figure 4) and Condition × Epoch [Wilks λ = 0.70, *F*(2,32) = 6.79, *p* = 0.003]. Greatest mean values were found during the acute caffeinated experimental phase and during the post-task epochs.

For RMSSD reactivity, two separate significant main effects were found for Condition and for Phase [Wilks λ = 0.70, *F*(1,33)= 13.92, *p* = 0.001] and [Wilks λ = 0.82, *F*(2,32) = 3.62, *p* = 0.038], respectively. A slight increase in reactivity was found during the acute caffeinated phase, whereas a small decrease appeared for the acute decaffeinated phase. A significant Condition × Task interaction was found. [Wilks λ = 0.89, *F*(1,33) = 4.28, *p* = 0.047]. The memory task was associated with increased RMSSD during the acute caffeinated period, whereas the opposite was true for the CP. In both conditions, overall positive differences between post-task and pre-task were found at Phase 2. 

### 3.4. Blood Pressure (BP)

Greatest DBP reactivity to tasks occurred at Phase 1, though no significant results were found for mean, reactivity, or recovery values of DBP. MANOVA tests revealed no significant effects for mean values of SBP. However, pairwise comparisons of mean SBP values revealed significant differences between Phases 1 and 2 (*p* < 0.00), as well as 2 and 3 (*p* < 0.00), but no significant differences between Phases 1 and 3 (*p* = 0.30).

No significant results were shown for SBP reactivity to tasks, though SBP increased in response to tasks at all time points during the caffeinated and decaffeinated days. Greatest SBP reactivity values appeared at Phase 2 following consumption of caffeinated coffee, whereas least SBP reactivity occurred during the acute phase following intake of decaffeinated coffee. For the difference between pre-task and post-task values for SBP, a marginal Condition × Task interaction was found [Wilks λ = 0.89, *F*(1,34) = 4.06, *p* = 0.052], whereby post-task values were closer to pre-task for the memory task versus the CP. This occurred to a greater extent on the caffeinated versus the decaffeinated day. 

Regarding mean arterial pressure (MAP), no significant results were found from a MANOVA test for either mean, reactivity, or recovery values. However, a pairwise comparison test revealed significant differences in mean MAP values between Phases 1 and 2 (*p* < 0.000), and 2 and 3 (*p* < 0.000), but no significant differences between Phases 1 and 3 (*p* = 0.052), with greatest values at Phase 2 (see Figure 5).

### 3.5. Pre-Ejection Period (PEP) and Left Ventricular Ejection Time (LVET) 

No significant caffeine-related effects were found for PEP. For mean LVET values, pairwise comparisons revealed no significant differences between mean PEP and LVET values at phases 1 and 3 (*p* = 1.00), but significant differences between phases 1 and 2 (*p* < 0.000), and between phases 2 and 3 (*p* < 0.000). No significant results were found for LVET reactivity.

### 3.6. Systemic Vascular Resistance (SVR) and Systemic Vascular Resistance Index (SVRI) 

A significant Condition × Phase × Epoch interaction was found for mean SVR [*F*(2.94,61.71) = 3.23, *p* = 0.029, η_p_^2^ = 0.133]. Greatest mean SVR occurred during Phase 2, in the caffeine condition. Mean SVR values during phases 1 and 3 did not significantly differ. Similarly, for mean SVRI values, a significant Condition × Phase × Epoch interaction was found [*F*(2.40,50.42), *p* = 021, η_p_^2^ = 0.156]. Greatest mean SVRI values occurred during tasks; however, in the caffeinated condition only, greatest mean SVRI values occurred at Phase 2. A significant Condition × Phase interaction was found for SVRI recovery [*F*(1.46,30.57) = 5.6, *p* = 0.015, η_p_^2^ = 0.211]. In the caffeine condition only, SVRI post-task epochs were significantly higher than pre-task values during Phase 2.

### 3.7. Self-Report Measures

On the Pleasant-Unpleasant dimension of the BMIS, a two-factor (Condition, Phase) repeated measures MANOVA revealed a significant main effect of Phase [Wilks λ = 0.73, *F*(2,37) = 6.87, *p* = 0.003]. Specifically, similar self-rated pleasantness occurred at Phase 1 on both days, but overall greater pleasant mood in the caffeine condition during Phases 2 and 3 as compared to the decaffeinated condition, wherein subjects reported overall decreases from Phase 1 to Phase 2 prior to increases at Phase 3. Although no significant main effect of Condition was found, indices of pleasantness were highest during the afternoon session on the caffeinated day. No significant main effects or interactions were found on the Arousal-Calm dimension. As expected, on the DASS, univariate repeated measures ANOVA revealed no significant main effect of Condition. Pearson correlations showed a significant positive relationship between the DASS Anxiety subscale and HF-HRV reactivity to the memory task at Phase 2 only on the caffeinated day (*r* = 0.38, *p* = 0.02). 

### 3.8. Ancillary Post Hoc Analyses

#### 3.8.1. Sex Differences

Partial correlations between acute caffeinated SBP and SVRI reactivity during tasks showed that for female subjects only, SBP and SVRI reactivity to the memory task were significantly negatively correlated (*r* = −0.56, *p* = 0.036). However, a significant positive correlation was shown for men (*r* = 0.63, *p* = 0.048). For acute caffeinated DBP and SVRI memory task reactivity, no significant relationship was found for women (*r* = −0.24, *p* = 0.238), whereas for men, a significant positive correlation was shown (*r* = 0.76, *p* = 0.014). No significant correlations were found for the CP task. Independent samples t-tests comparing acute caffeinated reactivity to tasks between male and female subjects for all other CV measures showed no significant mean differences. Multivariate tests revealed no significant sex differences in responses on either dimension of the BMIS, nor on the DASS.

#### 3.8.2. Task Ratings

Results from a two-factor repeated measures MANOVA revealed a significant main effect of Phase for difficulty ratings for the memory task [Wilks λ = 0.72, *F*(2,35) = 6.83, *p* = 0.003]. Greatest perceptions of difficulty occurred at Phase 1 on both the caffeinated and decaffeinated days. On both days, average ratings of difficulty decreased at Phase 2, with lowest ratings at Phase 3. Pain perception ratings for the CP task also revealed a significant main effect for Phase, whereby on average, greatest pain ratings occurred at Phase 1 and decreased across the measurement phases [Wilks λ = 0.82, *F*(2,34) = 3.75, *p* = 0.034]. Additionally, a significant interaction between Condition and Phase was found, with lowest pain ratings occurring at Phase 2 in the caffeine condition, but greatest ratings given at Phase 2 in the decaffeinated condition [Wilks λ = 0.78, *F*(2,34) = 4.81, *p* = 0.014] (see Figure 6).

#### 3.8.3. Consumption Patterns and Perceptions of Caffeine Condition

Subjects consumed a daily average of 309 mg of caffeine in the week preceding the study. Prior to the second day of the study, subjects consumed an average of 286 mg of caffeine daily. These averages include estimated amounts of caffeine from all sources. At debriefing, subjects provided a best guess about whether they had consumed caffeine on the first or second day of the study. Approximately half of subjects (48.7%) guessed correctly. They explained that their predictions were based on self-perceptions of comparative energy levels, memory task performance, taste of the coffee, and headaches on the decaffeinated day. 

## 4. Discussion

This is the first known study to investigate cardiovascular reactivity to caffeine with the specific aim of contributing empirically derived standards for experimental control for the drug. Our primary goal was to examine the efficacy of a six-hour abstention, based on the drug’s average half-life, in minimizing potentially confounding acute and negative abstention effects of caffeine on cardiovascular activity. As hypothesized, there was no evidence that cardiovascular activity or reactivity was elevated at the critical Phase 3 of the study. Similarly, no evidence suggested significant negative changes in mood or working memory task performance following six hours of deprivation. 

### 4.1. Physiological Responses

As predicted, results suggest that a six-hour abstention adequately controls for caffeine’s acute effects on BP and HR. While no systematic main effects or interactions appeared for mean BP or BP reactivity in response to caffeine, mean SBP and MAP values revealed the hypothesized pattern: relatively lower values during the first phase, greatest values during the acute caffeinated phase, followed by decreased values approximating the pre-caffeinated phase. Consistent with a large body of literature, caffeine consumption was associated with acutely increased BP and diminished HR, e.g., [13].

Lack of significant repeated measures MANOVA effects of caffeine on pressor reactivity may have resulted from highest mean BP reactivity during the first morning phase, rather than an absence of additive effects of caffeine and stress on BP. Regardless of condition, greatest overall pressor reactivity occurred during the morning sessions prior to coffee consumption and decreased sequentially across phases. Conceivably, greatest BP reactivity during the first experimental phase may have stemmed from novelty effects of the stressors, e.g., [45]. 

To address the relative lack of literature examining time-varying effects of caffeine on vagal activity, HF-HRV and RMSSD were acquired. Following effects of caffeine on HRV reactivity at Phase 2, there did not appear to be any enduring effects of the drug six hours after consumption. Results generally corroborate extant findings that acute caffeine ingestion may stimulate vagally mediated responses [2,46,47,48]. Small but significantly greater mean HF-HRV, RMSSD, and HF-HRV reactivity occurred on the caffeinated day. Although caffeine’s more reliable effects primarily influence sympathetic activity, moderate doses (e.g., 240 mg) can enhance modulation of parasympathetic activity [49]. Accompanying vagal stimulation could reflect a parasympathetic compensatory response. For example, recent research identifies a positive association between resting post-caffeinated BP and HF-HRV [50]. Results from the current investigation highlight the complexity of vagal and sympathetic activation related to caffeine use. 

Acute increases in HF-HRV in response to both tasks likely reflect effective stress regulation mechanisms. For example, both high-frequency power and sympathetic activity increase during cold stress due to subcutaneous thermoregulation [51]. Unexpected positive HF-HRV reactivity during the memory task may reflect adaptive emotion regulation strategies during the task [52]. Ostensibly, vagal fibers may be part of a negative feedback loop adjusting sympathoadrenal activity [53]. 

As anticipated for cardiac contractility indices, no significant effects of caffeine were found for PEP or LVET. Contrary to hypotheses, no consistent significant main effects of interactions were found for SVR or SVRI. However, SVRI values during post-task in Phase 2 significantly exceeded associated pre-task values in the caffeine condition only, suggesting an extended additive sympathetic effect of caffeine on cardiovascular activity, consistent with much literature suggests, e.g., [3,13]. Importantly, no significant differences in mean LVET, PEP, or SVR between pre-caffeine and afternoon experimental phases occurred.

An additional consideration regarding caffeine’s cardiovascular effects is sex differences in phasic caffeine-related reactivity to acute stressors. Sex differences for mean or reactivity magnitudes of BP or HR were not predicted in the present study. However, hypothesized sex differences in inotropic and chronotropic indices were partially corroborated. Despite similar degrees of caffeine-induced pressor changes among men and women, BP reactivity was positively associated with acute caffeinated SVRI reactivity for the memory task for men only. These results suggest differing mechanisms among men and women for acute caffeinated BP reactivity, reflecting similar findings from literature, e.g., [19]. Failure to find a consistent effect of acute caffeine consumption on SVR and SVRI may be partially due to sex differences. We did not examine cardiac output in this study; however, acute caffeinated BP reactivity to tasks may have been associated with greater increases in cardiac output among women compared to men. 

### 4.2. Self-Report Responses

No evidence of negative deprivation-related mood changes were present following the six-hour abstention period, corroborating overall hypotheses. Experimental phases were significantly associated with general mood indices as assessed by the BMIS. Along the Pleasant-Unpleasant dimension, greatest pleasant mood was reported during the afternoon session, particularly in the caffeinated condition. Caffeine’s association with positive affect is well supported in the literature, e.g., [54]. No significant Condition-or Phase-related differences along the Arousal-Calm dimension were reported. A positive relationship between the DASS Anxiety subscale and HF-HRV reactivity to the memory task during the acute caffeinated phase was unanticipated since state and trait anxiety are commonly associated with increased vagal withdrawal [21]. Further research must investigate caffeine’s potential modulatory effects on the relationship among stress, anxiety, and HF-HRV. 

Not surprisingly, self-reported task perception and autonomic reactivity suggest that the CP task was generally more aversive than the motivating memory task. The tasks also involve complex physiological, cognitive, and emotional responses that may markedly diverge. The incidence of highest pain rating at Phase 1 was expected. Some degree of nociceptive tolerance occurred over time, likely due to habituation mechanisms [55]. The CP task was least aversive during the acute caffeinated phase, suggesting the presence of known analgesic effects of caffeine, e.g., [56].

### 4.3. Limitations

Reduced sample size for ICG analyses reflects a limitation of the study. Additional potential limitations include self-reported caffeine abstention (versus biologically verified), practice effects on the memory stressor, and intrinsic individual differences in caffeine metabolism rates. Controlling for these inherent limitations was not feasible in this type of study design. Results from this investigation may not generalize to individuals with cardiovascular illnesses, younger or older populations, or those with vastly different caffeine consumption behavior. Given the particular sample selected, generalizability is somewhat intentionally limited. The sample selection allowed for increased comparability across a large number of studies conducted in university laboratories examining cardiovascular parameters, since college students are commonly recruited in this type of research. Caffeine consumption is widespread in this population, where 89% of participants in a large-scale study of college students reported past-30-day caffeine use and respondents were most likely to consume caffeine on a daily basis [57]. Other statistical approaches such as multilevel modeling may be useful in examining the issue of caffeine abstention effects on cardiovascular reactivity [58]. Finally, an attempt to control effects of circadian rhythms on cardiovascular parameters was made by completing the sessions at similar times across participants (i.e., morning sessions within two hours of each other and afternoon sessions within two hours of each other). 

As previously mentioned, a primary pharmacologic action of caffeine is its antagonism of adenosine receptors [13], which has sympathomimetic effects [8]. For this reason, control of caffeine is important in autonomic studies such as those that include cardiovascular variables. However, adenosine receptors are widely distributed in the brain, in particular areas in the basal forebrain [59] and various hypothalamic regions [60,61]. As such, caffeine has the potential to confound many types of variables, and in particular affect central nervous system (CNS) measures such as EEG and fMRI. We did not collect CNS data in this study, and so our findings cannot be directly extrapolated to them. However, we encourage the use of experimental designs such as the one used in this study n application to CNS measures, and so enhance experimental control in such research. 

## 5. Conclusions

Overall, results support the adequacy of a six-hour abstention duration prior to participation in studies of cardiovascular reactivity among young habitual consumers of moderate caffeine doses. This duration appeared adequate in reducing acute effects of caffeine on cardiovascular reactivity and circumventing caffeine deprivation effects. This conclusion is based on the absence of confounding effects of caffeine on cardiovascular reactivity during the afternoon session. The drug may have differing effects on task reactivity versus mean measures, (e.g., at caffeinated Phase 2, results showed significantly decreased mean HR, but increased HR reactivity). In addition to reactivity-related methodological concerns, caffeine’s actions on mean cardiovascular measures also present possible confounds in research by potentially producing ceiling effects. 

The current study is envisioned as a “proof of concept” study, and the first in a series of systematic investigations aiding in the formation of methodological standards by which to control for caffeine. Future research may examine effects at varying times of the day or using shorter abstention durations, which would be practically ideal. We designed this study with pragmatic concerns in mind; if six hours of abstention is an adequate control for caffeine, regular consumers could have a cup of coffee in the morning and participate in a study later that day. Not only would this buffer the effects of caffeine withdrawal, it could encourage participation among regular consumers. Future directions also include further examination of autonomic changes pertaining specifically to caffeine deprivation and withdrawal effects and the potential role of caffeine in acutely or more persistently altering vagal activity. In addition, replicating this study among various demographic groups (e.g., older populations) would expand knowledge of ideal caffeine control procedures in research using various subpopulations. Finally, it would be advantageous to explore the implementation of caffeine controls in clinical studies as well as other research studies beyond psychophysiological studies. 

The need to adequately and consistently control for caffeine in cardiovascular research is evident. Abstention effects manifest both physiologically and psychologically, and may pose as much of a threat to experimental validity as caffeine itself. Potential confounds associated with caffeine are likely to partially underlie widely variable results in research examining caffeine’s effects on psychophysiological indices. The present study contributes to enhancing reliability and validity in psychophysiological research by suggesting a control standard for the world’s most commonly consumed psychoactive substance.

## Figures and Tables

**Figure 1 brainsci-13-00224-f001:**
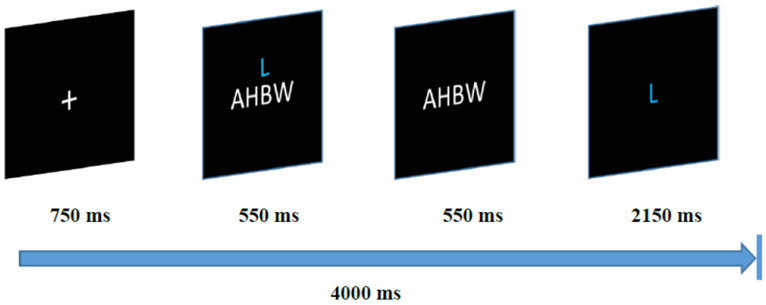
Working Memory Task. Note. Each trial consists of presentation of a fixation cross on a computer screen for 750 ms followed by a series of nonsense letters consisting of four capital letters presented for 550 ms. The letter series was then masked, with a single blue letter appearing above the masked series for the remainder of the trial. Subjects must then press one of two keys indicating whether or not the blue letter was presented in the preceding letter series. Each trial was a total of 4000 ms long. Subjects completed 14 practice trials before beginning the 3 min task. Subjects were instructed to attempt to respond correctly for at least 90% of 44 total trials for a cash bonus [35,36].

**Figure 2 brainsci-13-00224-f002:**
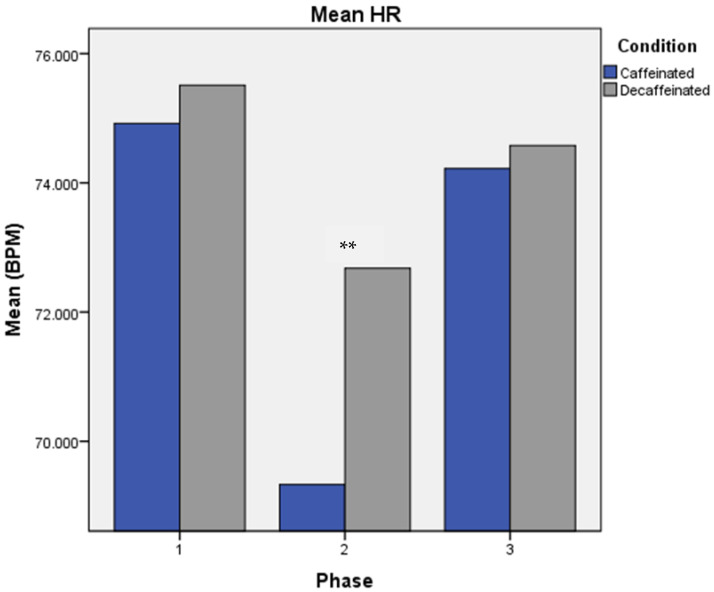
Mean Heart Rate (HR), Condition × Phase. Note. Error bars are omitted from graphs due to inherent inaccuracy in reporting error for repeated measures MANOVA results in SPSS. Experimental phases were as follows: Phase 1: measures taken prior to consumption of coffee. Phase 2: acute measures collected 30 min following intake of coffee, and Phase 3: measures collected six hours following consumption of coffee. ** *p* < 0.05.

**Figure 3 brainsci-13-00224-f003:**
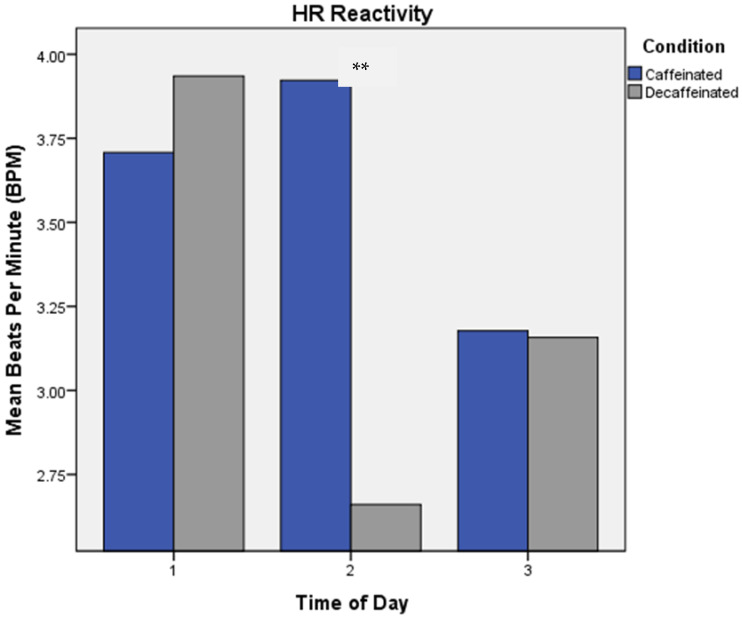
Mean HR Reactivity by Phase. Experimental phases were as follows: Phase 1: measures taken prior to consumption of coffee. Phase 2: acute measures collected 30 min following intake of coffee, and Phase 3: measures collected six hours following consumption of coffee. ** *p* < 0.05.

**Figure 4 brainsci-13-00224-f004:**
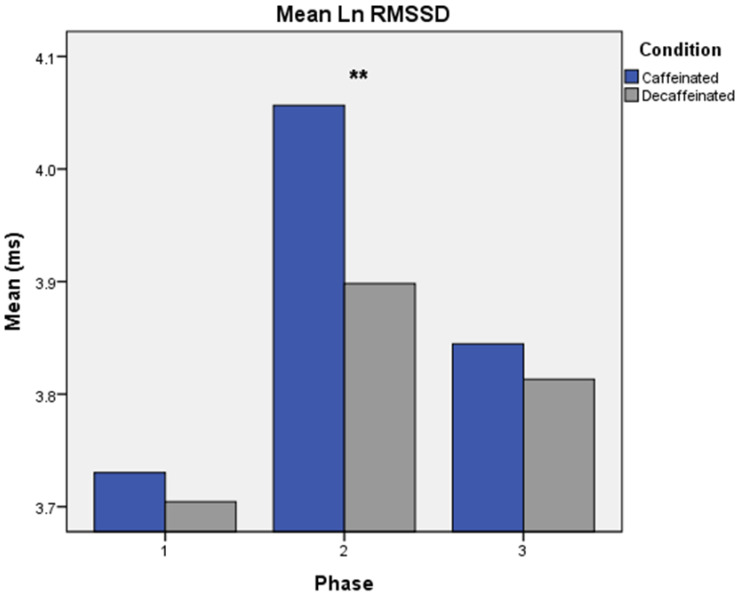
Mean RMSSD, Condition × Phase. Note. Condition × Phase interaction. Experimental phases were as follows: Phase 1: measures taken prior to consumption of coffee. Phase 2: acute measures collected 30 min following intake of coffee, and Phase 3: measures collected six hours following consumption of coffee. ** *p* < 0.05.

**Figure 5 brainsci-13-00224-f005:**
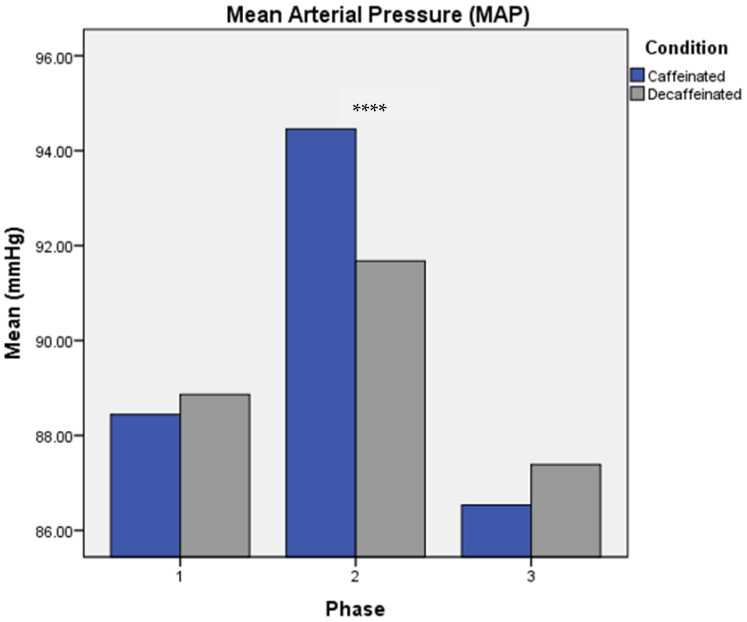
Pairwise Comparisons for Mean MAP. *Note*: Significant differences between Phases 1 and 2 (*p* < 0.000), and 2 and 3 (*p* < 0.000), but no significant differences between Phases 1 and 3 (*p* = 0.052). Main effect of Phase significant at *p* = 0.000. Experimental phases were as follows: Phase 1: measures taken prior to consumption of coffee. Phase 2: acute measures collected 30 min following intake of coffee, and Phase 3: measures collected six hours following consumption of coffee. **** *p* < 0.000.

**Figure 6 brainsci-13-00224-f006:**
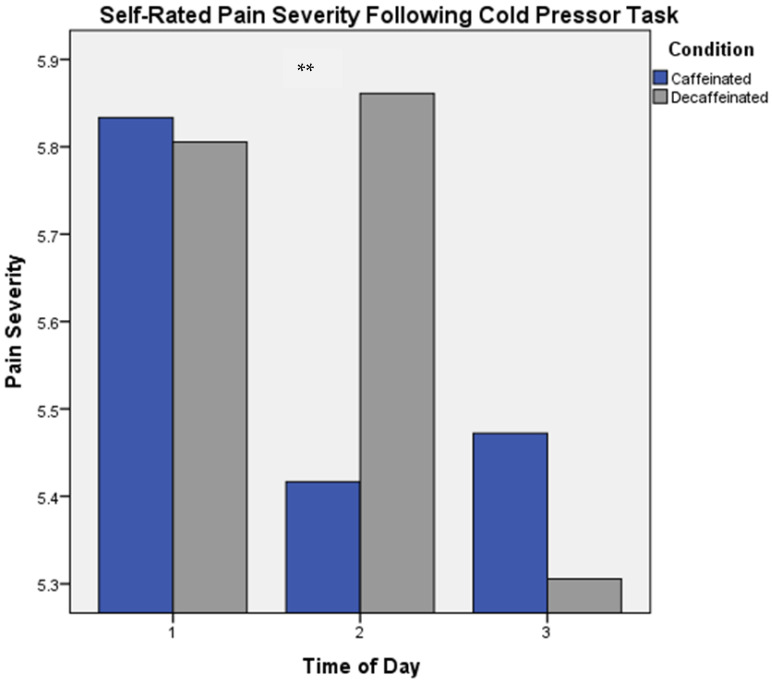
Self-rated Pain Severity for Cold Pressor, Condition × Phase. Experimental phases were as follows: Phase 1: measures taken prior to consumption of coffee. Phase 2: acute measures collected 30 min following intake of coffee, and Phase 3: measures collected six hours following consumption of coffee. ** *p* < 0.05.

**Table 1 brainsci-13-00224-t001:** Descriptive statistics.

Demographic Measures	Mean ± SD or N (%)
Age (years)	20.9 ± 1.92
Race	
Caucasian African American/Black Asian Latino/Hispanic	26 (60.4%) 1 (2.3%) 11 (25.5%) 1 (2.3%)
BMI (kg/m^2^)	
Male Female	25.9 ± 3.43 24.0 ± 3.52
Baseline Physiological Measures HR (bpm) RR (bpm) Ln HF-HRV (ms^2^) Ln RMSSD (ms) SBP (mmHg) DBP (mmHg) MAP (mmHg) PEP (ms) LVET (ms) SVR (dynes·s·cm^−5^) SVRI (dynes·s·cm^−5^·m^2^)	Mean (SE) 72.43 (1.76) 16.38 (0.54) 6.63 (0.17) 3.70 (0.11) 120.42 (1.64) 69.50 (1.09) 86.76 (1.22) 96.52 (2.40) 340.60 (4.14) 1038.61 (45.76) 1973.65 (102.43)

**Table 3 brainsci-13-00224-t003:** Mean Physiological Values Stratified by Task and Condition for Phase 2 (Acute Drug Phase).

Physiological Measure	Mean (SD)	Mean (SD)	N
**Caffeinated Day**	Memory Task	Cold Pressor Task	
HR (bpm)	70.19 (13.96)	73.22 (11.45)	36
RR (bpm)	18.63 (3.51)	16.10 (4.59)	39
Ln HF-HRV (ms^2^)	7.43 (1.09)	7.16 (1.25)	36
Ln RMSSD (ms)	4.14 (0.59)	4.00 (0.50)	36
SBP (mmHg)	131.14 (14.45)	137.32 (17.93)	37
DBP (mmHg)	77.42 (9.67)	83.09 (11.88)	37
MAP (mmHg)	95.74 (10.82)	101.02 (15.64)	37
PEP (ms)	99.84 (15.90)	99.15 (15.44)	24
LVET (ms)	357.02 (27.67)	358.84 (23.54)	24
SVR (dynes·s·cm^−5^)	1124.85 (298.87)	1186.24 (271.25)	24
SVRI (dynes·s·cm^−5^·m^2^)	2124.77 (737.86)	2224.13 (667.94)	24
**Decaffeinated Day**			
HR (bpm)	73.78 (12.38)	75.50 (10.62)	36
RR (bpm)	18.06 (3.25)	16.44 (4.05)	39
Ln HF-HRV (ms^2^)	7.01 (1.08)	6.82 (1.25)	36
Ln RMSSD (ms)	3.89 (0.54)	3.88 (0.58)	36
SBP (mmHg)	126.52 (12.95)	138.77 (14.86)	37
DBP (mmHg)	73.53 (9.48)	82.45 (10.69)	37
MAP (mmHg)	91.35 (10.36)	100.34 (12.60)	37
PEP (ms)	101.21 (13.53)	98.54 (10.77)	24
LVET (ms)	347.33 (27.65)	357.07 (25.72)	24
SVR (dynes·s·cm^−5^)	1188.92 (426.12)	1059.03 (282.11)	24
SVRI (dynes·s·cm^−5^·m^2^)	2236.12 (990.19)	1956.95 (672.61)	24

Note. HR = heart rate; RR = respiration rate; HF-HRV = high frequency heart rate variability; RMSSD = root mean square successive differences; SBP = systolic blood pressure; DBP = diastolic blood pressure; MAP = mean arterial pressure; PEP = pre-ejection period; LVET = left-ventricular ejection time; SVR = systemic vascular resistance; SVRI = systemic vascular resistance index.

**Table 4 brainsci-13-00224-t004:** Mean Reactivity on Caffeinated Day, by Experimental Phase and Task.

Cardiovascular Measure	Experimental Phase	Mean Reactivity (± SD)	
		Memory Task	CP Task
HR (bpm)	1 2 3	3.88 (5.80) 3.06 (7.24) 3.77 (5.73)	3.54 (5.72) 4.79 (6.74) 2.58 (0.77)
Ln HF-HRV (ms^2^)	1 2 3	0.14 (0.90) 0.43 (1.01) 0.26 (0.75)	0.29 (0.76) 0.03 (0.73) 0.27 (0.77)
Ln RMSSD (ms)	1 2 3	0.05 (0.32) 0.11 (0.39) 0.09 (0.36)	0.06 (0.36) −0.06 (0.28) 0.06 (0.30)
SBP (mmHg)	1 2 3	5.69 (6.98) 3.14 (6.11) 3.73 (5.64)	13.98 (8.22) 11.97 (10.29) 11.84 (7.42)
DBP (mmHg)	1 2 3	3.41 (5.20) 2.39 (5.62) 2.88 (4.20)	11.34 (6.30) 10.41 (7.66) 7.69 (6.21)
MAP (mmHg)	1 2 3	4.29 (5.57) 2.84 (5.34) 3.41 (4.51)	12.76 (7.62) 10.24 (11.35) 9.98 (7.42)
PEP (ms)	1 2 3	1.87 (8.28) −0.35 (4.37) 1.26(5.91)	−0.02 (8.92) −0.52 (6.41) 1.80 (5.78)
LVET (ms)	1 2 3	−0.25 (10.41) 0.04 (11.18) 0.69 (11.81)	8.45 (18.29) 5.95 (14.48) 7.32 (16.97)
SVR (dynes·s·cm^−5^)	1 2 3	−8.94 (89.90) −24.36 (77.91) 11.76 (56.93)	58.48 (140.89) 51.18 (282.74) 69.73 (151.14)
SVRI (dynes·s·cm^−5^·m^2^)	1 2 3	−15.62 (174.26) −55.47(165.93) 21.78 (108.40)	97.50 (275.38) 46.67 (665.42) 126.34 (283.47)

Note. Reactivity values were calculated by subtracting baseline from task values. Experimental phases were as follows: Phase 1: measures taken prior to consumption of coffee. Phase 2: acute measures collected 30 min following intake of coffee, and Phase 3: measures collected six hours following consumption of coffee.

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
