# Peer review of "How Long Is Long Enough? Controlling for Acute Caffeine Intake in Cardiovascular Research"

_brainsci, 2023, doi:10.3390/brainsci13020224_

Round 1

Reviewer 1 Report

This is a very interesting paper investigating the effects of caffeine on cardiovascular outcomes, highlighting the need to control caffeine consumption in cardiovascular research. The paper is well written and if of interest for the readers. However, several minor changes should be recommended to improve the paper.

Introduction.

1- The introduction section is too long.

The authors are firstly introducing the caffeine use in the general population, why is important to study, and then, they are introducing the physiology of caffeine. 1.1. subsection is not needed.

2- The subsection 1.2.Current study, can be renamed as "Aims and objectives". The authors can also draw thew hypotheses in this section.

Methods.

1-The subsection 2.1. Subjects, can be renamed as Participants and study design.

2-The main procedures of the paper are described in 2.2. subsection. I recommend to divide this subsection into several stages of the study.

3-I recommend to build a table presenting all the measures used in the study: 1) Physiological measures, 2) Psychological measures, 3) experimental tasks.

The results are presented in a good manner.

Discussion

1-Why is self-report responses described separated in the discussion section?

To expand the section about future directions is recommended: clinical, research (animal and human studies).

Reviewer 2 Report

The article deals with a very important and as yet undeveloped topic - the question of experimental control for the drug. The authors suggested that 6 hours is the optimal period without caffeine to conduct subject-controlled studies.

The literature analysis covers a sufficient number of sources, the authors consistently consider all necessary logical components.

The authors call their study "proof of concept." Indeed, this design can also be applied to finding a time interval to control consumption of other substances (e.g., alcohol).

Very importantly, in this study, caffeine conditions were counterbalanced across morning sessions and subjects were 272 blind to the condition assigned.

The authors are very detailed about the design of the experiment, but I could not find detailed information about the psychophysiological measurements. How exactly were they conducted and with what duration? For frequency and some statistical analysis on heart rhythm metrics the duration of the recording and its quality is very important.

Round 2

Reviewer 1 Report

The authors have worked hard on the comments and have substantially improved the paper. They have followed all suggestions and reorganized the paper according to the recommendations about sections, subsections, etc.

I consider they have made a great work.

Reviewer 2 Report

I agree with the current version.